# Humor Styles, Bullying Victimization and Psychological School Adjustment: Mediation, Moderation and Person-Oriented Analyses

**DOI:** 10.3390/ijerph191811415

**Published:** 2022-09-10

**Authors:** Christoph Burger

**Affiliations:** 1Department of Developmental and Educational Psychology, Faculty of Psychology, University of Vienna, Liebiggasse 5, A-1010 Vienna, Austria; christoph.burger@univie.ac.at; Tel.: +43-1-293-1532; 2Department of Cognition, Emotion, and Methods in Psychology, Faculty of Psychology, University of Vienna, Liebiggasse 5, A-1010 Vienna, Austria; 3Division of Psychological Methodology, Department of Psychology and Psychodynamics, Karl Landsteiner University for Health Sciences, Dr.-Karl-Dorrek-Straße 30, A-3500 Krems, Austria

**Keywords:** bullying, school violence, victimization, bully-victims, humor styles, psychological adjustment, school adjustment, affiliative humor style, self-defeating humor style, aggressive humor style

## Abstract

Humor can be both adaptive and maladaptive and plays a role in bullying victimization and school adjustment. It was hypothesized that humor styles decrease or increase victimization, which in turn affects school adjustment. Furthermore, humor might moderate effects of victimization on school adjustment. Moreover, a person-oriented approach could improve our understanding of group differences in these variables. An online questionnaire retrospectively surveyed emerging adults (*N* = 172; 77.2% female; mean age: 22.7 years) with respect to humor style use, bullying victimization and school adjustment. Mediation and moderation analyses were computed, and two sets of person-oriented analyses compared victims, bully-victims and noninvolved students on humor styles and school adjustment, and three latent humor-related groups (overall-high, adaptive-high and adaptive-low) on victimization and school adjustment. Victimization fully mediated the positive effect of affiliative humor and partially mediated the negative effect of self-defeating humor on school adjustment. The negative effect of victimization on school adjustment was magnified by self-defeating humor and attenuated by aggressive humor. Bully-victims used both aggressive and self-defeating humor more frequently, and victims used aggressive and affiliative humor less frequently. Furthermore, both victims and bully-victims showed lower school adjustment. Finally, the adaptive-high humor group showed lower victimization and higher school adjustment. Implications for school interventions are discussed.

## 1. Introduction

Bullying in schools poses a global threat to the healthy development of students, as it is often associated with serious and potentially long-lasting consequences for the students involved [1]. Both bullying victimization and bullying perpetration were found to affect not only student health and well-being but also psychological adjustment to the demands of the school environment [2]. Previous research has shown that the use of different humor styles, which can be adaptive or maladaptive, might play an important role in social situations and might have an effect on psychological adjustment and school success [3]. The use of different humor styles might influence the likelihood of becoming a victim of bullying. In addition, when used as coping strategies, different humor styles might worsen or neutralize the negative influence of ongoing bullying victimization with respect to school adjustment. Such effects could also take the form of group differences in humor use and school adjustment between bullying-related groups (such as victims, bully-victims or noninvolved students) or of group differences in victimization and school adjustment between student groups who use specific humor styles profiles. Unfortunately, there is a dearth of studies empirically examining the interplay of these factors. The present study aims to fill this knowledge gap by conducting both variable-oriented analyses to investigate mediation and moderation effects and person-oriented analyses to examine differences between bullying- and humor-related groups.

During school years, individuals are continuously presented with new social and academic challenges to which they have to adjust [4]. They also gain more independence from their parents or guardians, and interactions with peers become increasingly important in their daily lives. These interactions are opportunities to make new friends but are also potential sources of interpersonal problems such as bullying. Because the use of humor plays an important role in social interaction at school, it is essential not to overlook the role of humor when examining the relationship between bullying victimization and school adjustment [5].

### 1.1. Humor Styles

The use of humor is a stable character trait that people of all ages constantly demonstrate in their social interactions [6]. It influences the quality of relationships with others and plays a role in all aspects of life, not only in everyday life and at work, but also at school. Children with a good sense of humor, for example, have been shown to perform better in social and academic situations [3]. Humor has also been repeatedly identified as a stress-reducing coping strategy [7] as it may allow individuals to partially detach themselves psychologically from their immediate circumstances and to view situations from a fresh and less threatening perspective. It can also be used to defuse and reduce conflict by softening tensions and pointing out ambiguities in a face-saving way [6]. Humor also has a dark side. It can be used to make fun of and marginalize others and to express hostility and aggression [5]. These darker aspects of humor, such as irony, sarcasm and cynicism, are associated with psychopathy [8]. Due to this complexity, there is little agreement in the extant literature on how to define humor. However, there is a consensus that humor is not a single dimension but a multi-layered construct. A widely used and accepted conceptualization of humor by Martin et al. [9] divides humor into four broad styles based on the intersection of the two dimensions of being benign vs. harmful and of aiming to increase one’s self-worth vs. the worth one has to others (see Figure 1): (i) affiliative, (ii) self-enhancing, (iii) aggressive and (iv) self-defeating. Previous research has shown that these humor styles have differential effects on psychological well-being and psychological adjustment [10,11].

The affiliative humor style is described as using humor for the purpose of becoming closer to others by elevating their mood and well-being [12]. The goal is to become more attractive to peers by entertaining them with humor that is benevolent and accepting towards others [13]. This form of humor may not only strengthen existing friendships and facilitate finding new friends but also can help resolve social conflicts in school more easily. Two recent meta-analyses found affiliative humor to be positively associated with mental health and subjective well-being [10,11].

The self-enhancing humor style is also a benign form of humor. Users of this style attempt to support and empower themselves by employing humor as a stress management strategy or defense mechanism [14]. In this manner, they can distance themselves from adverse circumstances and rise above them in a humorous way without disparaging others. Using self-enhancing humor is associated with lower levels of neuroticism [15], can help students deal with stress at school and other social challenges and contribute to higher levels of school adjustment [16] by buffering against any negative effects on psychological health and well-being [17]. Thus, similarly to the affiliative humor style, self-enhancing humor is positively related to mental health and subjective well-being [10,11].

The aggressive humor style, unlike the two styles already discussed, is not benevolent and refers to the hostile use of humor at the expense of other people and groups and is associated with actual aggressive behavior [18]. Individuals who use aggressive humor may pretend to be playfully funny but try to make themselves look better by aggressively belittling or teasing others. The use of aggressive humor can be perceived as hostile and lead to conflict escalation, which may be socially undesirable and can jeopardize existing relationships with classmates [19]. Users of this humor style tend to report harboring more negative feelings towards school [3]. Aggressive forms of humor can be associated with psychopathy and sadism [20]. In two recent meta-analyses, aggressive humor style was found to be overall unrelated to mental health but to damage subjective well-being [10,11].

Finally, the self-defeating humor style refers to disparaging oneself in a humorous way in front of others in order to gain their recognition at one’s own expense [21]. This can be described as a misdirected attempt at stress reduction that involves auto-aggressive components and the repression of one’s actual emotional needs [22]. Using this humor style has been linked to higher levels of neuroticism [15] and psychological adjustment problems [23]; more negative feelings toward school [3]; higher levels of loneliness, depression, anxiety and psychiatric symptoms; and lower levels of self-esteem [24,25]). In two recent meta-analyses, self-defeating humor was negatively related to both mental health and subjective well-being [10,11].

Summarizing, humor styles have been found to be associated with psychological adjustment [26]. Self-enhancing and affiliative humor styles are considered positive and adaptive individual traits that may protect against psychological adjustment problems by buffering against negative effects. Aggressive and self-defeating humor styles, on the other hand, are considered to be maladaptive strategies and act as risk factors. The use of humor styles may affect how well students can function and adapt to the demands of school [3].

### 1.2. School Bullying

School bullying is characterized by repeated and systematic malicious harassment over an extended time period inflicted on a victim that is less powerful than the perpetrator or group of perpetrators [27]. Bullying results in serious consequences for students’ overall mental health and in psychological adjustment problems [2]. In terms of school adjustment, students who are victims of bullying tend to have lower levels of school attachment [28], perform worse in school [29], stay away from school more often [30] and dislike being in school because they feel unsafe [31]. Unlike normal peer conflicts in which neither side has a power advantage, bullying is persistent and tends to intensify over time [32]. It is typical for victims of bullying that they cannot escape this vicious circle of victimization on their own and depend on help from others who are outside the negative behavioral peer dynamics (e.g., teachers [33]). It is therefore reasonable to assume that while using humor can help students resolve normal conflicts, it may be ineffective in the context of ongoing bullying victimization [34].

It is, however, plausible that in the lead-up to potential bullying victimization, the use of humor styles may influence the likelihood that victimization will actually occur. Specific humor styles might be antecedent risk or protective factors in the process leading up to bullying victimization. Bullying perpetrators have, for example, shown to choose vulnerable students as their victims [35] because they might not fight back. Students using self-defeating humor could be signaling vulnerability and insecurity to potential perpetrators, thereby increasing the chance of victimization. Students who use self-enhancing humor, on the other hand, might leave a stronger and less vulnerable impression on potential perpetrators [18] because such a humor style allows them to appear more nonchalant and emotionally stable in social interactions [17]. In addition, students using aggressive humor, by its confrontational nature, may repeatedly insult and provoke classmates and increase the risk for conflict escalation. In the long run, they may have only few or no friends who might protect them. This unpopularity and lack of inclusion in the peer group may render them more likely to be the target of bullying [36]. In contrast, student who use affiliative humor might experience higher popularity and peer acceptance [37], which could lead to a larger network of friends and peer supporters who come to their aid if they are targeted and work together to help them stand up against the perpetrator. Previous research indeed found that all four humor styles are associated with peer victimization [38]. Self-defeating humor was most strongly positively associated and affiliative humor was most strongly negatively associated with bullying victimization [36,38].

In response to victimization, general humor use was found to be a promising and effective coping strategy [39], especially among male students [40], and led to higher resilience in victims [41] and fewer depressive symptoms in boys with high negative emotionality following peer aggression [7]. Different humor styles may be able to reduce or worsen (i.e., moderate) negative effects of bullying victimization on psychological school adjustment. Previous research suggests that some victims use self-defeating humor as a misguided attempt to be liked more by and become closer to their perpetrators [34]. As a maladaptive coping strategy, self-defeating humor is very likely to exacerbate negative effects on school adjustment. Aggressive humor could make victims’ situations worse, as it could be perceived as a provocation by the perpetrators, which could lead to a further escalation of the victimization by the perpetrators [34] and to lower school adjustment. Self-enhancing humor, through its effect of empowering victims and distancing them from psychological distress, could act as a buffer, thus acting as a positive resource that reduces the negative effects of victimization on school adjustment. Finally, the use of affiliative humor might buffer against the negative effects because victims can maintain some level of social standing despite victimization, which allows them to maintain some level of self-dignity and mental health.

### 1.3. Complementing Variable- with Person-Oriented Analyses

The current dominant paradigm in bullying research is the variable-oriented approach. This research approach focuses on variables that capture specific constructs across individuals and examines associations between these variables at the level of the overall sample. Although this approach has contributed substantially to the understanding of psychological phenomena, it also has its limitations because understanding human phenomena is ultimately not about variables but about individuals who are to be understood as integrated wholes [42]. These limitations of the variable-oriented approach can be overcome by complementing it with the person-oriented approach, which focuses on group memberships and typologies that arise from the integration of several variables (i.e., variable profiles).

Thus, variable-oriented analyses of victimization should be complemented by person-oriented approaches examining differences among different bullying-related groups [2]. While the variable-oriented approach relates the victimization variable to other variables across the entire sample in the form of models of varying degrees of complexity (e.g., bivariate correlations, mediation, or moderation models), the person-oriented approach focuses on subsamples of individuals and their membership to specific bullying-related groups and, for example, examine differences between these groups. It makes sense to distinguish between person-oriented groups such as noninvolved students, “pure“ victims who do not tend to be proactively aggressive and bully-victims who in addition to playing the role of victims also play the role of perpetrators [43]. This differentiation might be interesting because one of the humor styles, aggressive humor, seems to be associated with bullying perpetration, especially when the humor moves away from “good-natured banter” to the other end of the continuum: abuse [44]. The use of aggressive humor might be attractive to those who bully because they can hide their negative intentions behind supposedly well-intentioned jokes and justify their behavior by alluding to their humorous character [20]. Bully-victims might also react with aggression in response to rejection by peers [45]. It is therefore plausible that bully-victims use different humor styles than victims or noninvolved students (e.g., higher levels of aggressive humor style). Another reason why it is interesting to study bully-victims as a separate group is that previous studies have identified them as an even greater risk group for psychological adjustment problems than the pure victim group [46], and regarding school adjustment, bully-victims generally have higher absenteeism from school than victims and report less support from teachers [47].

Similarly, the variable-oriented examination of the effects of humor styles has recently been characterized as reductionist, and it has been recommended that effects of humor styles should be complemented by person-oriented approaches that analyze typical humor style configurations (i.e., profiles of concurrent humor styles use) rather than single humor styles [48]. For example, individuals scoring high on self-defeating humor and low on the other three humor styles were identified to have the lowest score of psychological adjustment compared to other humor style profiles [23]. When directly compared with humor styles, humor profiles were shown to have consistently greater predictive value for friendship and well-being outcomes [48]. Therefore, in addition to the variable-oriented analyses of individual humor styles (in the form of mediation and moderation models), the current study will also examine person-oriented differences between latent profiles of combined humor style use.

### 1.4. Short-Term Retrospective Measurement of Bullying-Related Behavior

Eliciting ongoing bullying behavior from students comes with several potential problems. If they are currently involved in bullying, they might be reluctant to report being victimized because they might be ashamed of it or, in the case of bullying perpetration, might fear punishment [2]. Moreover, asking them about their bullying-related behaviors could elicit negative feelings from them [49], which might raise ethical concerns and possible validity problems. These problems can be circumvented by retrospective measurement by young adults who have recently completed their school, because reporting unwanted behaviors might be less self-threatening due to the temporal distance, and more pronounced recall biases can be prevented because the time elapsed is not so long. The advantages and disadvantages of this approach have been described elsewhere [2].

### 1.5. Current Study

Previous research indicates that humor has an important role in the interplay between bullying and psychological school adjustment. To date, however, only a limited number of studies have been conducted in this domain. Thus far, no study has examined the extent to which specific humor styles predict bullying victimization and whether the effects of different humor styles on school adjustment are mediated by victimization. Furthermore, the extent to which humor styles mitigate or worsen the negative effects of victimization on school adjustment is unknown. This study aims to address these research gaps by pursuing the following four objectives.

The first goal is to explore the mediating role of bullying victimization on the effect of humor styles on school adjustment: in other words, whether the use of certain humor styles is related to increased or decreased victimization and, thus, in a further step, may encourage or discourage psychological school adjustment. It is hypothesized that affiliative and self-enhancing humor styles are positively associated with school adjustment while aggressive and self-defeating humor styles are negatively associated with school adjustment and that these effects are at least partially mediated by bullying victimization (i.e., significant indirect effects).

Second, the study hypothesizes that victimization is negatively correlated with school adjustment and explores whether this effect is moderated by the use of different humor styles. It is hypothesized that high levels of affiliative and self-enhancing humor styles decrease this negative effect (i.e., less steep decline in the slope) and that high levels of aggressive and self-defeating humor styles renders this effect even more negative (i.e., more steep decline in the slope).

Third, the study takes a person-oriented approach, aiming to compare noninvolved students with victims and bully-victims (i.e., students who are both victims and perpetrators). It is hypothesized that victims have lower levels of self-enhancing and affiliative humor and higher levels of self-defeating humor than noninvolved students. It is further hypothesized that in addition to the effects hypothesized for victims, bully-victims also show higher levels of aggressive humor.

Finally, again from a person-oriented perspective, the study seeks to explore which latent profile groups of concurrent humor style use were present in the current sample and to analyze group differences between them. Since the nature of the latent profiles found is not known in advance, no explicit hypotheses were formulated.

## 2. Materials and Methods

### 2.1. Participants

A total of 278 German-speaking adolescents accessed an online questionnaire. A minimum age of 18 was set as the inclusion criterion, as participants should have completed their school years in order to be able to assess them retrospectively in their entirety. After entering their demographic information, 11 individuals who did not meet this criterion were thanked and informed that they were not part of the target group. Furthermore, 86 participants dropped out during the process of filling out the questionnaire (they completed on average *M*_dropout_ = 37.07% of the questionnaire, *SD* = 20.93) and were therefore excluded from the final data set. These participants did not differ in terms of gender (*P*_dropout_female_ = 78.8%; *P*_final_final_ = 77.2%, Χ^2^(1) = 0.087, *p* = 0.77) but were slightly younger (*M*_dropout_age_ = 22.07, *SD* = 1.96) than those in the final sample (*t*(258) = 2.19, *p* = 0.03, *d* = 0.29).

The final sample consisted of 172 individuals (77.2% female) with a mean age of 22.70 years (*SD* = 2.29, *Min* = 18, *Max* = 27). Regarding their occupational status (multiple selection was possible), 79.7% reported being a university student, 45.9% being employed, 1.7% reported being unemployed, 1.7% reported being in military or civilian service and 0.6% reported being a housewife/househusband. Regarding the highest level of education completed, 63.4% reported having obtained a secondary school diploma (equivalent to general university entrance qualification), 29.7% of participants reported having obtained a bachelor’s degree, 4.1% reported having obtained a master’s degree and 2.9% reported having completed vocational training (including apprenticeship or vocational school).

### 2.2. Procedure

Online data collection was conducted cross-sectionally at one point in time and was advertised via online social networks. Participants were informed about the main facts of the survey on the first questionnaire page (e.g., inclusion criteria, topic areas queried, expected duration, voluntariness of participation, data protection, expected benefits and risks) and had to provide informed consent to complete the questionnaire. Participants were not remunerated. On each page of the online questionnaire, participants were clearly informed that the questions referred to school years and were to be answered retrospectively to the best of their recollection.

### 2.3. Measures

After providing informed consent, participants were asked to indicate their gender (0 = female, 1 = male), age and highest previous education completed. Moreover, the following four constructs were measured. Reliabilities, means, standard deviations, and bivariate zero-order correlations of the main study variables are presented in Table 1.

#### 2.3.1. Use of Humor Styles

The humor styles questionnaire (HSQ) [13] was translated into German and adapted in order to retrospectively measure four different humor styles used in the school context with other classmates during the school years. In order to keep the overall questionnaire short, a selection of 18 7-point Likert items was provided (the selection was made using corrected item–total correlation coefficients available from a previous study, with details omitted). Answer options ranged from *totally disagree* (1) to *totally agree* (7). The four humor styles measured are labeled affiliative (4 items), aggressive (5 items), self-enhancing (4 items) and self-defeating (5 items). Example items are “When other adolescents made a mistake, I often teased them about it” and “I did not have to try very hard to make other adolescents laugh—I was a naturally funny person at school”.

#### 2.3.2. Use of Humor in Conflict Situations

The scale by Smith et al. [50] was translated into German and adapted to retrospectively measure humor use in conflict situations during the school years. It consisted of 3 5-point Likert items with answer options ranging from *strongly disagree* (1) to *strongly agree* (5). An example item is “I joked and laughed in everyday school life to play down the seriousness of a disagreement”.

#### 2.3.3. Psychological School Adjustment

Nine 7-point Likert items by Burger and Bachmann [2] were used to capture psychological school adjustment. The items were adapted to refer retrospectively to the school years [2]. Answer options ranged from *completely disagree* (1) to *completely agree* (7). Sample items are “When I was in school, I was happy” and “When I was in school, I worried”. After scoring the items, higher values represented higher levels of adjustment.

#### 2.3.4. Bullying Victimization

The following item was used in order to gauge whether individuals were victimized: “I was harassed, picked on in my class”. Answer options were *No, that does not apply to me* (1), *Yes, sometimes* (2) and *Yes, often* (3). When participants answered with answer option 2 or 3, the 5-item 7-point measure by Burger and Bachmann [2] was used to measure bullying victimization [2]. Participants retrospectively reported how they were targeted by five different bullying behaviors during their school years, including physical, verbal, relational, property-related and cyber forms of bullying. Answer options ranged from *strongly disagree* (1) to *strongly agree* (7). Before answering the items, participants read a brief instruction, explaining the retrospective measure and that it refers to their school years. The items were prefaced with “How can you describe the harassment in more detail?”. Sample items were as follows: “It happened with words (I was called names, yelled at, laughed at, etc.)”, “It happened physically (I was pushed, kicked and punched)” and “It happened with the help of the Internet (rumors were spread online, videos of me were sent online against my will, etc.)”. A factor analysis with principal component extraction indicated a one-factor solution explaining 67.16% of variance.

In order to be able to distinguish between victims and bully-victims, those who indicated that they were victimized were also asked a further question regarding bullying perpetration, which was also answered on a 7-point scale, ranging from *strongly disagree* (1) to *strongly agree* (7) [2]. The item reads “I myself have also teased (one or more) other classmate/s who were inferior to me for an extended period of time, with the intention of making that person suffer”.

### 2.4. Missing Data

Percentages of missing values were 1.7% for gender, 0.0% for age and 0.6% for class conflict frequency. The maximum percentage of missing values was 0.0% across the 18 variables measuring humor styles (affiliative, aggressive, self-enhancing and self-defeating), 0.0% across the three variables measuring humor use in conflicts, 1.2% across the nine variables measuring psychological school adjustment, and 1.7% across the five variables measuring bullying victimization.

### 2.5. Data Analytical Strategy

As a first step, bivariate associations between study variables were calculated before including them in more complex models.

#### 2.5.1. Mediation Analysis

A mediation model was calculated using statistics program JASP version 0.14.1.0 [51]. Humor styles were included as predictors, bullying victimization as the mediator, and psychological school adjustment as the outcome, while controlling for gender, age and classroom conflict frequency. A full information maximum likelihood estimator was used that could handle missing values. Direct, indirect, total indirect and total effects were calculated.

#### 2.5.2. Moderation Analysis

A moderated multiple regression analysis was calculated with IBM SPSS Statistics 27 using the PROCESS macro [52]. The predictor was bullying victimization, and the outcome variable was psychological school adjustment. Humor styles were included as moderator variables and gender, age and classroom conflict frequency were included as covariates. All predictors were mean-centered, and the interaction terms were calculated from mean-centered predictors.

#### 2.5.3. Determining Bullying-Related Groups

Using a person-oriented approach, bullying-related groups were formed. Following the procedure used by Kollerová et al. [53], all individuals who scored at least 0.5 *SD*s above the overall sample mean score in bullying victimization were assigned to a temporary victimization group. Next, individuals within this temporary group were identified who also reported having committed bullying perpetration. Finally, mutually exclusive bullying-related groups were formed, representing noninvolved individuals, pure victims and bully-victims. As a next step, descriptive statistics and prevalence percentages were calculated for these three groups.

#### 2.5.4. Differences between Bullying-Related Groups

A series of ANCOVAs was conducted using the statistics program JASP version 0.14.1.0 [51] to test for differences between noninvolved individuals, pure victims and bully-victims on different humor-related variables and psychological school adjustment.

#### 2.5.5. Determining Latent Profiles of Humor-Related Groups

Using statistical software Jamovi [54] with module snowRMM [55] and R package tidyLPA [56], a set of latent profile analyses was carried out, testing the fit indices of models with 2, 3, 4 and 5 latent classes.

#### 2.5.6. Differences between Humor-Related Latent Groups

To better understand and describe the found latent profile classes, a set of four ANCOVAs was conducted with each humor style as the dependent variable and the latent classes as independent variable. Gender, age and classroom conflict frequency were included as covariates. Furthermore, a series of ANCOVAs was conducted to test for differences between the three humor-related groups regarding humor use in conflicts, victimization and psychological school adjustment.

## 3. Results

### 3.1. Bivariate Correlations of Study Variables

Bivariate relations among the main study variables are shown in Table 1. Being male was associated with higher levels of aggressive humor. Age was positively associated with self-enhancing humor. A higher conflict frequency in the school class was positively associated with being a pure bully, with being a bully-victim, with higher levels of victimization, with higher levels of self-defeating humor, and with lower levels of psychological school adjustment. Being a victim and being a bully-victim were both positively correlated to victimization. Victimization was correlated to higher levels of self-defeating humor. However, being a victim was associated with lower levels of aggressive humor, and being a bully-victim was associated with higher levels of both aggressive and self-defeating humor. Almost all humor styles intercorrelated positively with small to large effect sizes (exceptions: aggressive humor style did not correlate with affiliative nor with self-enhancing humor). Humor use in conflicts was correlated positively with all humor styles. Psychological school adjustment was positively correlated to self-enhancing humor and negatively correlated to being a victim, being a bully-victim, victimization and self-defeating humor.

### 3.2. Mediation Model: Does Victimization Mediate the Effect of Humor Styles on School Adjustment?

A mediation model with multiple predictors and covariates (see Figure 2) was used to determine path coefficients effects (Appendix A) and total, direct and indirect effects (Table 2). The model revealed significant positive total effects for affiliative (γ = 0.172, *p* = 0.03), self-enhancing (γ = 0.301, *p* < 0.001) and self-defeating (γ = −0.453, *p* < 0.001) humor on psychological school adjustment. The effect of affiliative humor on school adjustment was fully mediated by bullying victimization (indirect effect: γ = 0.073, *p* < 0.05), and the effect of self-defeating humor on school adjustment was partially mediated by bullying victimization (indirect effect: γ = −0.115, *p* < 0.01; direct effect: γ = −0.338, *p* < 0.001). The effect of self-enhancing humor was not mediated by bullying victimization (direct effect: γ = 0.289, *p* < 0.001). Aggressive humor showed neither significant direct nor indirect effects on school adjustment.

### 3.3. Moderation Model: Can Humor Styles Dampen or Strengthen the Negative Association between Victimization and School Adjustment?

A moderated regression model predicting psychological school adjustment with victimization as predictor and humor styles as moderators and gender, age and class conflict frequency as covariates was calculated using the PROCESS macro [52] (see Table 3). All predictors and covariates that define interaction terms were mean-centered to ease interpretability. A heteroscedasticity consistent standard error and covariance matrix estimator was used (Huber–White). The model was significant (*F*(12, 154) = 19.727, *p* < 0.001) and explained 50.03% of variance of psychological school adjustment.

Positive conditional main effects were found for affiliative and self-enhancing humor, and negative conditional main effects were found for self-defeating humor. For participants with average levels of victimization and with average levels of humor styles (except for the described predictor) with all other predictors being equal, affiliative and self-enhancing humor styles were positively associated with psychological school adjustment (*b*_affiliative_ = 0.166, *p* = 0.019; *b*_self-enhancing_ = 0.270, *p* < 0.001), whereas self-defeating humor style was negatively associated with psychological school adjustment (*b*_self-defeating_ = −0.364, *p* < 0.001). For participants with average levels of humor styles with all other predictors being equal, victimization was negatively associated with psychological school adjustment (*b*_victimization_ = −0.298, *p* < 0.001).

Furthermore, the effect of victimization on school adjustment was significantly moderated by aggressive and self-defeating humor. The inclusion of the interaction between victimization and aggressive humor style significantly accounted for further 2.3% of explained variance (*F*(1, 154) = 6.137, *p* = 0.014), and the inclusion of the interaction between victimization and self-defeating humor style explained an additional 3.1% of variance (*F*(1, 154) = 12.689, *p* < 0.001).

For participants with low and average use of aggressive humor (77.25% of participants), victimization was negatively associated with school adjustment, whereas there was no significant association between victimization and school adjustment for participants with high levels of aggressive humor (above the score of 3.82; 22.75% of participants). Aggressive humor seemed to act as a buffer, dampening the negative effect of victimization. A visual representation of the interaction effect of aggressive humor is shown in Figure 3. The Johnson–Neyman technique for probing conditional effects for different values of the moderator aggressive humor showed that victimization negatively predicts school adjustment only for low values of aggressive humor up to a value of 3.82 and becomes non-significant thereafter (see Appendix A).

For participants with average or high use of self-defeating humor (76.65% of participants), victimization was negatively associated with school adjustment, whereas there was no significant association between victimization and school adjustment for participants with low levels of self-defeating humor (below the score of 2.38; 23.35% of participants). Self-defeating seemed to be a risk factor, strengthening the negative effect of victimization. A visual representation of the interaction effect is shown in Figure 4. The Johnson–Neyman output describing conditional effects for different values of self-defeating humor showed that victimization negatively predicts school adjustment only for high values of self-defeating humor down to a value of 2.38 and becomes non-significant thereafter (see Appendix A).

### 3.4. Results Regarding Bullying-Related Groups

#### 3.4.1. Determining Bullying-Related Group Membership

Following the procedure used by Kollerová et al. [53], all individuals who scored at least 0.5 *SD* above the overall sample mean score in bullying victimization were assigned to a temporary victimization group (*n* = 44). Next, victims were identified who reported also carrying out bullying perpetration (*n* = 21).

Finally, the following mutually exclusive bullying groups were formed (see Appendix A): (1) 74.0% noninvolved, (2) 13.6% victims and (3) 12.4% bully-victims. The three groups did not differ significantly regarding gender distribution nor regarding mean age. Regarding the mean frequency of classroom conflicts, noninvolved students (*M_noninvolved_* = 2.74, *SE* = 0.08) reported lower levels than both victims (*M_victims_* = 3.39, *SE* = 0.18, *M_diff_* = 0.66, *SE_diff_* = 0.20, *t*(2) = 3.33, *p_tukey_* = 0.003, Cohen’s *d* = 0.78) and bully-victims (*M_bully-victims_* = 3.43, *SE* = 0.19, *M_diff_* = 0.69, *SE_diff_* = 0.21, *t*(2) =3.39, *p_tukey_* = 0.003, Cohen’s *d* = 0.84).

#### 3.4.2. Bullying-Related Group Differences in Humor Styles

A series of ANCOVAs (see Table 4 and Figure 5) revealed significant effects of bullying-related behavioral roles on *aggressive*, *self-defeating* and *affiliative* humor style after controlling for gender, age and conflict frequency in class. Tukey post hoc tests on the *aggressive* humor style revealed significant mean differences between victims (*M_adj_* = 2.54, *SE* = 0.24) and the other two groups: noninvolved students (*M_adj_* = 3.23, *SE* = 0.11; *M_diff_* = 0.69; *SE_diff_* = 0.25, *t*(2) = 2.72, *p* = 0.020, *d* = 0.64) and bully-victims (*M_adj_* = 3.71, *SE* = 0.25; *M_diff_* = 1.18; *SE_diff_* = 0.33, *t*(2) = 3.56, *p* = 0.001, *d* = 1.15). A further Tukey post hoc test on the *self-defeating* humor style revealed significant mean differences between noninvolved students (*M_adj_* = 3.08, *SE* = 0.12) and bully-victims (*M_adj_* = 3.84, *SE* = 0.28; *M_diff_* = 0.77; *SE_diff_* = 0.30, *t*(2) = 2.58, *p* = 0.029, *d* = 0.64). Finally, a Tukey post hoc test on the *affiliative* humor style revealed marginally significant mean differences between noninvolved students (*M_adj_* = 5.55, *SE* = 0.12) and victims (*M_adj_* = 4.96, *SE* = 0.26; *M_diff_* = 0.59; *SE_diff_* = 0.28, *t*(2) = 2.13, *p* = 0.09, *d* = 0.51). This effect is considered to be relevant, since the results were significant at the multivariate level.

#### 3.4.3. Bullying-Related Group Differences in Psychological School Adjustment

A series of ANCOVAs (see Table 4 and Figure 5) revealed significant effects of bullying-related behavioral roles on *psychological school adjustment* after controlling for gender, age and conflict frequency in class. Tukey post hoc tests on *psychological school adjustment* revealed significant mean differences between noninvolved students (*M_adj_* = 5.64, *SE* = 0.12) and both of the other groups: victims (*M_adj_* = 4.08, *SE* = 0.25, *M_diff_* = 1.56; *SE_diff_* = 0.27, *t*(2) = 5.82, *p* < 0.001, *d* = 1.44) and bully-victims (*M_adj_* = 4.70, *SE* = 0.27, *M_diff_* = 0.95; *SE_diff_* = 0.28, *t*(2) = 3.38, *p* = 0.003, *d* = 0.84).

### 3.5. Results Regarding Humor-Related Latent Profile Groups

#### 3.5.1. Determining Humor-Related Group Membership

A set of four latent profile analyses ranging from two to five latent classes were computed. A three-class solution was chosen because the bootstrapped likelihood ratio test indicated a significant increase in model fit between the two-class and the three-class model; adding a fourth or a fifth class, however, did not significantly increase model’s fit (see Appendix A for fit indices of all models and Figure 6 for a latent profile plot of the three-class model). The three-class model also had the best (=smallest) BIC (although the AIC was slightly smaller in the five-class solution) and the highest minimum of the average latent class probabilities and acceptable entropy.

Class 1 included 22.1% of participants (*n* = 38), class 2 included 50.0% of participants (*n* = 86) and class 3 included 27.9% of participants (*n* = 48). To compare classes across each humor style, a set of four ANCOVAs was performed. The covariates were gender, age and class conflict frequency. Each humor style differed across the three classes (see Appendix A). Tukey post hoc tests for each humor styles across the classes were conducted. Class 1 had the highest scores in almost all humor styles compared to the other classes, with the only exception being aggressive humor, which only differed marginally (*p* = 0.058) from class 2. We thus termed class 1 *overall high*. Class 2 had higher scores than class 3 regarding self-enhancing and affiliative humor but did not differ regarding aggressive and self-defeating humor. We thus termed class 2 *adaptive high*. Class 3 had the significantly lowest scores in self-enhancing and affiliative humor compared to the other groups. We thus termed class 3 *adaptive low*.

#### 3.5.2. Humor-Related Group Differences in General Humor Use in Conflicts, Bullying Victimization and Psychological School Adjustment

An ANCOVA was calculated with general humor use in conflicts as outcome variable. There was a main effect for the latent humor profile classes (see Table 5 and Figure 7). Tukey post hoc tests showed that all three groups differed significantly from another, with the “overall high” group exhibiting the highest values and “adaptive low” exhibiting the lowest values (overall high vs. adaptive high: *M_overallhigh_adj_* = 4.22, *SE* = 0.125; *M_adaptivehigh_adj_* = 3.69, *SE* = 0.089; *M_diff_* = 0.53; *SE_diff_* = 0.148, *t*(2) = 3.59, *p* = 0.001, *d* = 0.712; overall high vs. adaptive low: *M_overallhigh_adj_* = 4.22, *SE* = 0.125; *M_adaptivelow_adj_* = 3.33, *SE* = 0.117; *M_diff_* = 0.90; *SE_diff_* = 0.165, *t*(2) = 5.42, *p* < 0.001, *d* = 1.31; adaptive high vs. adaptive low: *M_adaptivehigh_adj_* = 3.69, *SE* = 0.089; *M_adaptivelow_adj_* = 3.33, *SE* = 0.117; *M_diff_* = 0.36; *SE_diff_* = 0.135, *t*(2) = 2.70, *p* = 0.021, *d* = 0.49).

A further ANCOVA (see Table 5 and Figure 7) revealed marginally significant effects of humor-related latent profile groups on *bullying victimization* after controlling for gender, age and classroom conflict frequency. Tukey post hoc tests, however, revealed that the class “adaptive high” (*M_adj_* = 1.63, *SE* = 0.160) had a significantly lower victimization score than the class “adaptive low” (*M_adj_* = 2.21, *SE* = 0.210; *M_diff_* = 0.59; *SE_diff_* = 0.242, *t*(2) = 2.42, *p* = 0.044, *d* = 0.42). All other post hoc tests were non-significant.

A final ANCOVA (see Table 5 and Figure 7) identified significant effects of humor-related latent profile groups on *psychological school adjustment* after controlling for gender, age and classroom conflict frequency. Tukey post hoc tests revealed that the class ”adaptive high” (*M_adj_* = 5.67, *SE* = 0.147) had a higher school adjustment score than the other two classes “overall high” (*M_adj_* = 5.07, *SE* = 0.205; *M_diff_* = 0.60; *SE_diff_* = 0.243, *t*(2) = 2.47, *p* = 0.038, *d* = 0.47) and “adaptive low” (*M_adj_* = 4.90, *SE* = 0.193; *M_diff_* = 0.77; *SE_diff_* = 0.222, *t*(2) = 3.48, *p* = 0.002, *d* = 0.66). The groups “overall high” and “adaptive low” did not differ significantly.

## 4. Discussion

Previous studies indicate that humor plays an important role in the interplay between bullying and psychological school adjustment. However, to date, there has been a dearth of studies in this area. No study has examined the extent to which the use of specific humor styles predicts bullying victimization and whether the effects of different humor styles on school adjustment are mediated by victimization. Furthermore, the extent to which different humor styles moderate the negative effects of victimization on school adjustment is unknown. The current study aims to close these gaps. Since person-oriented group comparisons often provide complementary information to variable-oriented analyses, this study also compared different bullying-related groups (victims, bully-victims, noninvolved students) and different latent humor-related groups (overall high, adaptive high and adaptive low) on relevant study variables.

The most important findings of this study relate to the insight that including the interplay between humor styles and victimization is important when seeking to gain a deeper understanding of the mechanisms behind school adjustment. Victimization can be a powerful mediator variable in the relationship between some humor styles (e.g., affiliative humor) and school adjustment. In addition, different methods of using humor as a coping strategy can be effective in breaking or strengthening the negative link between victimization and school adjustment. Person-oriented analyses showed that noninvolved students, victims and bully-victims differ in terms of humor-related variables and school adjustment, and that different latent humor class profiles differ in terms of humor-related variables, victimization and school adjustment.

### 4.1. Adverse Effects of Self-Defeating Humor

In the mediation model, self-defeating humor style was a risk factor, having a negative total effect on school adjustment. This negative effect was partly mediated by victimization, suggesting that, in addition to the direct negative effect, higher levels of self-defeating humor led to higher levels of victimization, which subsequently led to poorer school adjustment. In the moderation model, self-defeating humor was identified as maladaptive coping strategy. It moderated the negative effect of victimization on school adjustment, suggesting that higher levels of self-defeating humor would exacerbate the negative effect.

All these findings are in line with previous studies showing that self-defeating humor use is a maladaptive humor style linked to harmful effects for its users. The silver lining to these findings is that reducing self-defeating humor use could effectively reduce both victimization and school adjustment problems and neutralize the negative effects of victimization on school adjustment. However, why is this humor style still used by students? One reason may be that students believe that by making fun of themselves, they can avoid appearing arrogant or off-putting to others, which may allow them to build bridges to others more easily [12]. It could also be that these students are limited to using self-defeating humor because adaptive humor styles require greater interpersonal skills, which they may never have learned to use successfully [12]. In victimization cases, victims may use self-defeating humor as a misguided attempt to mirror and please the perpetrators in the hope that this will cause them to stop the bullying. Students who do not want to escalate their own victimization further may attempt to defuse the situation with a self-depreciating “survival” strategy of expressing subordination by lowering their own status and of making clear that they do not want to provoke the perpetrator in any way. Furthermore, joining in with the perpetrators’ hurtful remarks can also be regarded as a pre-emptive strategy to increase the feeling of control [57] or as a concealing strategy by the victims to make themselves appear less vulnerable by masking their insecurities and their distress [25].

While there is evidence that self-defeating strategies may well yield positive results for high-status individuals [57] who might be relatively safe from bullying victimization, it may backfire for low-status individuals by making them less attractive for social interaction and resulting in the development of maladaptive social support networks [58]. For potential perpetrators in particular, this submissive behavior may create the impression that these students lack the ability to assert and defend themselves, making them optimal victims [36]. In ongoing victimization scenarios, the use of self-defeating humor may provide some short-term relief, but in the long run, perpetrators may reinforce bullying behavior because they realize they can get away with anything without expecting any resistance from victims. Using this strategy might increase the victims’ load of humiliation in various ways. On top of the distress caused by the public humiliation inflicted by the perpetrators, self-defeating humor might add further distress caused by the self-inflicted public humiliation. The victims’ public self-deprecation may not only validate the perpetrators’ behavior but may also be interpreted by other peers as an implicit endorsement by the victims such that other peers may feel invited to join in the bullying. The fact that victims have to suppress their actual feelings when using this humor style could also contribute to greater distress.

### 4.2. Beneficial Effects of Aggressive Humor

In the mediation model, the aggressive humor style was identified as neither a risk nor a protective factor for students using this humor style. It did neither predict victimization nor school adjustment. In the moderation model, however, aggressive humor was identified as a protective coping strategy. It moderated the negative effect of victimization on school adjustment, suggesting that higher levels of aggressive humor would neutralize the negative effect whereas low levels would exacerbate it.

The initial assumption that using aggressive humor would be associated with greater victimization as a result of increased interpersonal conflicts and a diminished social network could not be confirmed. The findings are, however, in line with previous research showing that aggressive humor style use does not necessarily hurt mental health (Schneider et al. [11]; but see also Jiang et al. [10]). It is plausible that students who use aggressive humor are not attractive victims in the eyes of the perpetrators, as they seem assertive and might fight back. Using aggressive humor might also be associated with a stress-buffering effect that reduces psychological distress [59]. A major caveat, however, is that aggressive humor has been found to be associated with bullying perpetration [60,61] in previous studies and, thus, might be a major factor in promoting victimization in those who are the targets of the humor [12].

### 4.3. Beneficial Effects of Affiliative Humor Style

In the mediation model, affiliative humor style was found to be a protective factor, associated with higher levels of school adjustment. This positive effect was fully mediated by bullying victimization, suggesting that the positive effect of affiliative humor on school adjustment is entirely driven by reducing victimization experiences. In the moderation model, affiliative humor did not moderate the effect of victimization on school adjustment.

These results are in line with previous research showing that victimization is associated with lower scores in affiliative humor style [36,38] and with higher levels of mental health and subjective well-being [10,11] and school success [3]. Still, it is surprising that affiliative humor has no direct positive effects on school adjustment. It is an important extension of existing knowledge of the underlying mechanisms that the positive effect of affiliative humor on school adjustment is entirely mediated via the path of the prevention of or reduction in peer victimization. It is plausible that students who are proficient in the use of affiliative humor may facilitate acceptance and popularity among peers, which may have positive effects on their social network by helping in maintaining their friendships and making new friends [12]. In addition, there could be a self-reinforcing effect in which students who are well-accepted by their peers may also benefit from a more comfortable social environment in which they have more opportunities to improve their affiliative humor skills [37]. The underlying mechanism could therefore be that students with a large network of friends are less likely to be selected as victims of perpetrators because perpetrators face more resistance and support from the victims’ social networks [58].

### 4.4. Beneficial Effects of Self-Enhancing Humor Style

In the mediation model, self-enhancing humor style was identified as a protective factor because it had a positive direct effect on school adjustment. It was, however, not associated with bullying victimization. In the moderation model, self-enhancing humor did not moderate the negative impact of victimization on school adjustment.

The positive effects of self-enhancing humor style are in line with previous research [10,11]. However, the hypothesis that the use of self-enhancing humor leads to less victimization by increasing mental health (e.g., self-esteem) and assertiveness in behavior or that it can mitigate the negative effects of victimization was not supported. It is important to note that self-enhancing humor, although not protective against victimization or the effects of victimization, is nonetheless an adaptive and beneficial factor for increasing school adjustment.

### 4.5. Person-Oriented Group Comparisons

The person-oriented analyses comparing bullying-related groups were largely consistent with the results of the variable-oriented analyses. Disadvantageous group membership was linked to higher levels of self-defeating humor (i.e., bully-victims had higher scores than noninvolved students) and lower levels of affiliative humor (i.e., victims had lower scores than noninvolved students). Being a victim was linked to lower aggressive humor use (i.e., compared to both noninvolved students and bully-victims). Unexpectedly, we found no significant differences in the use of aggressive humor between noninvolved students and bully-victims, as we expected bully-victims to employ more aggressive humor in order to bully other more vulnerable students as a way of coping with the negative effects of their own victimization experiences [62], thereby further spreading victimization experiences among classmates. It is possible that the smaller number of people in the bully-victims group may contribute to this phenomenon by reducing statistical power. In line with variable-oriented results, there were no group differences regarding self-enhancing humor style use. Interestingly, noninvolved students, victims and bully-victims did not differ in the frequency with which they used humor in conflict situations, suggesting that the effects depended not on the quantity but on the type of humor. Finally, both victims and bully-victims experienced poorer academic adjustment than noninvolved students, which is in line with previous research [2].

The person-oriented latent humor profile analysis revealed that half (about 50%) of the students used high levels of adaptive humor styles and low levels of maladaptive humor styles (“adaptive high”). More than a quarter of students used low levels of both adaptive and maladaptive humor styles (“adaptive low”), and less than a quarter of students used high levels of all humor styles (“overall high”). All groups differed significantly regarding humor use in conflict situations, with the overall high group having the highest score, being followed by the adaptive high group and the adaptive low group. These results show that student groups differ in their nuanced use of humor styles (indiscriminate use of all styles vs. nuanced use or non-use of adaptive styles). As expected, having an adaptive humor profile was linked to less victimization (compared to the adaptive low group) and to higher levels of school adjustment (compared to both the overall high and the adaptive low group). This confirms that the latent humor style profiles found in the sample meaningfully differ in important outcome variables. Future studies should further investigate the explanatory potential of this group distinction.

### 4.6. Practical Implications

The results of this study have implications for schools and in particular for teachers, who are at the forefront of the fight against bullying [63]. It could be that some students use maladaptive humor styles because they have never learned to use adaptive styles and, therefore, lack these interpersonal competencies [12]. It has been proposed that teaching about humor and how to use it skillfully could be a useful anti-bullying intervention [40]. In particular, schools could be places where students learn adaptive ways of using humor by creating a classroom that embraces adaptive forms (i.e., affiliative and self-enhancing) and discouraging maladaptive (i.e., self-defeating and aggressive) forms of humor [64]. Teachers can also model the constructive use of humor using model learning. Although the use of aggressive humor appears to protect against the victimization of those who use it, this type of humor should not be recommended as it has been shown to lead to higher perpetration by those who use it and, thus, higher victimization of other students [60], especially in online settings [61]. It is important that teachers know that using aggressive humor can represent bullying perpetration, which can have serious effects on the targets. For teachers, the use of aggressive humor on the part of students might present a challenge because it is often ambiguous [65], and the line between banter (which may also be used between group members to show affection and intimacy) and harmful perpetration is not easily discernible [61] and might be exploited by perpetrators hiding their malicious intentions behind humorous statements [20]. Previous studies have shown that victims who also take part in perpetration (i.e., bully-victims) are a particularly difficult group for teacher interventions to reach [66]. Classmates can also play a vital role as socializing agents [67] and should be empowered to recognize maladaptive humor use (i.e., the negative impact of self-defeating humor; bullying disguised as well-intentioned aggressive humor) and how to intervene in bullying situations [68], although potential iatrogenic effects of peer defending have not yet been well researched [69].

Finally, more direct school intervention programs have been developed to increase students’ awareness of different humor styles and their effects, and their use has resulted in students becoming more aware of and better able to consider the consequences of their use of different humor styles, which has the potential to positively impact their social development [70]. It should not be overlooked that a change in humor style can also include work on emotion regulation skills. The ability to effectively regulate both anger [71] and characterological self-blame seems to be particularly important [72], as these may be associated with the use of maladaptive humor styles.

### 4.7. Limitations and Future Directions

In addition to the strengths of the study such as the application of a mediation and a moderation model and the combination of variable- and person-oriented approaches, the current study is not free from limitations. Among these, one limitation was the cross-sectional design of the study. It is clear that the models, although jointly analyzing multiple variables, are simplified accounts of reality as bidirectional effects could be at work, since the use of humor styles and school adjustment could be both predictors and consequences of bullying victimization [26,38]. Next, the sample consisted of German-speaking adolescents with a relatively high level of education (about one third had an academic degree); therefore, caution should be exercised when applying results to other demographic groups. Despite social media penetration is close to 100 percent among Austrian youth [73], the fact that an online survey was used and promoted on online social networks may have skewed the results, as individuals who are less active on social networks had a reduced chance in the study’s participation. Furthermore, the current study focused on bullying victimization. In order to identify bully-victims, only individuals who reported victimization were also asked to provide information about their taking part in bullying perpetration. In future, it might be of interest to investigate bullying perpetration across the entire sample in order to identify and examine “pure” bullies (who are not victims) and to examine possible associations with humor style use. In addition, because victims and bully-victims are narrower target populations, bullying-related subgroups were smaller in size, and person-based comparisons had less than optimal statistical power.

Future research efforts should replicate the present results in large-scale longitudinal studies with students of different ages and from different socio-cultural contexts. In addition, it may be valuable to further examine the impact of social desirability in retrospective self-reports of adverse experiences (such as victimization). Supplementing self-reports with peer or teacher reports and observation-based measures of bullying experiences could also contribute to alleviating such biases [74].

## 5. Conclusions

The school years are a period with many challenges, such as finding a place in the social fabric of the school class and resolving conflicts with peers as constructively as possible. Unfortunately, when bullying occurs, the victims’ abilities to stop it are per definition extremely limited. It is beneficial to avoid potential victimization by other classmates as much as possible before it even occurs and to deal with victimization as adroitly as possible through the appropriate use of coping strategies to prevent serious and long-term effects on mental health and academic performance at school. This study has shown that the use of humor styles can play an important role in these efforts. Self-enhancing humor styles had direct positive effects on school adjustment, whereas self-defeating humor styles showed a direct negative effect. Using high levels of affiliative humor style or low levels of self-defeating humor decreased victimization and, in turn, increased school adjustment. Furthermore, when coping with victimization, using low levels of self-defeating humor and high levels of aggressive humor could potentially eliminate the negative effects of victimization on school adjustment. The use of adaptive humor profiles is associated with lower levels of bullying victimization and higher levels of school adjustment. The findings have important practical implications for teachers and school intervention programs seeking to improve students’ humor style usage. First and foremost, efforts should be made to increase both affiliative and self-enhancing humor in students and to reduce self-defeating humor. Aggressive humor, despite its buffering effect, should not be recommended because it may increase the level of hostility in the classroom and may spread negative effects of bullying to other more vulnerable classmates rather than completely eliminating them.

## Figures and Tables

**Figure 1 ijerph-19-11415-f001:**
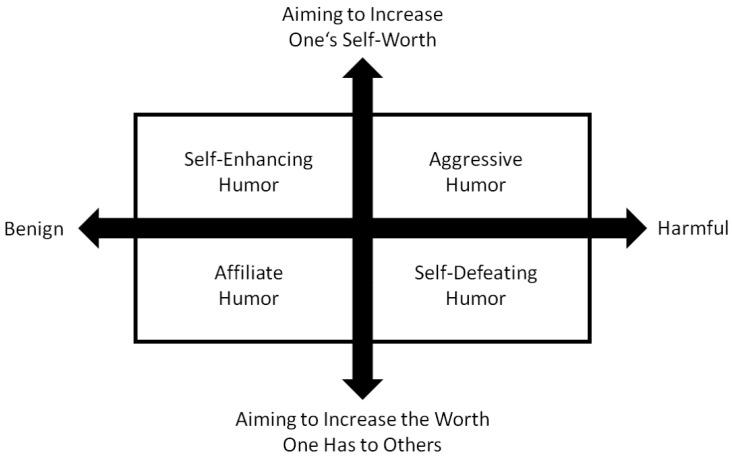
Model of the four humor styles and the intersection of the two dimensions *benign* vs. *harmful* and *aiming to increase the self-worth* vs. *aiming to increase the worth one has to others*.

**Figure 2 ijerph-19-11415-f002:**
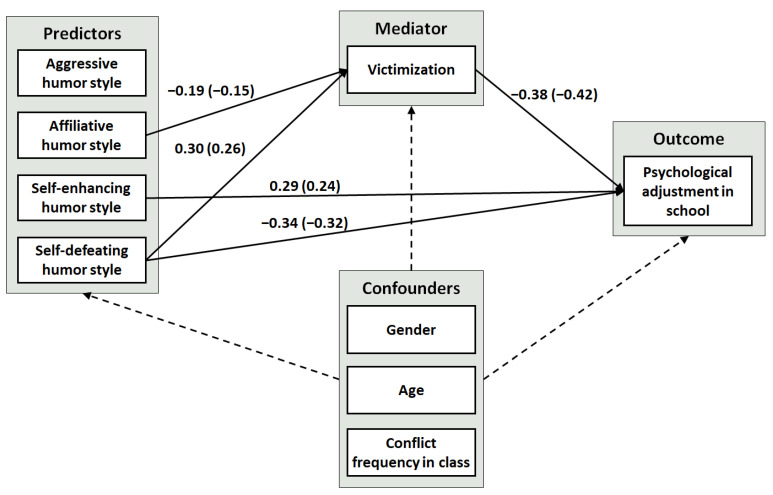
Mediation model: The effect of humor styles on psychological school adjustment with bullying victimization as the mediator variable. *Note.* Mediation analysis was calculated with JASP [51]. Only significant paths and loadings (*p* ≤ 0.05) are presented. Loadings are unstandardized coefficients with standardized coefficients in parentheses.

**Figure 3 ijerph-19-11415-f003:**
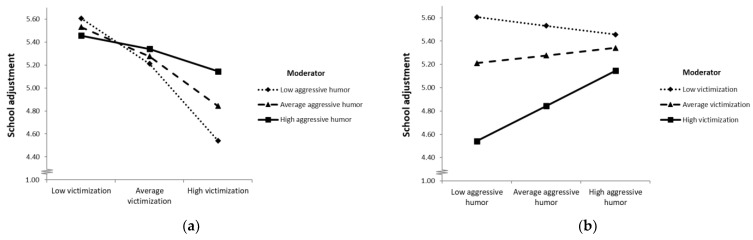
Profile plot of the hybrid interaction effect of aggressive humor style and victimization on school adjustment. (**a**) Psychological school adjustment (plotted on the vertical axis) as a function of victimization (low, average and high; plotted on the horizontal axis) for different levels of aggressive humor style use (low, average and high; plotted as separate lines). (**b**) Psychological school adjustment (plotted on the vertical axis) as a function of aggressive humor style use (low, average and high; plotted on the horizontal axis) for different levels of victimization (low, average and high; plotted as separate lines). *Note*. Data for visualizing the conditional effects were taken from the syntax output of the PROCESS macro v4.00 [52]. Values for victimization (low = 1.00; average = 1.86; high = 3.31); values for aggressive humor style (low = 1.91; average = 3.04; high = 4.16).

**Figure 4 ijerph-19-11415-f004:**
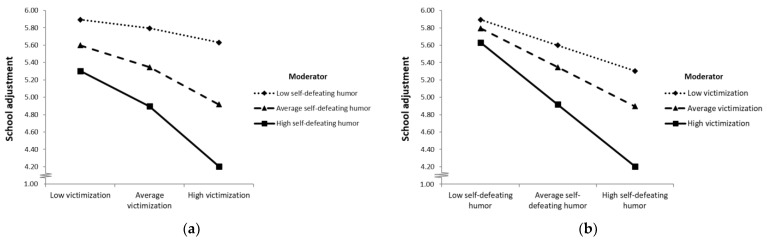
Profile plot of the ordinal interaction effect of self-defeating humor style and victimization on school adjustment. (**a**) Psychological school adjustment (plotted on the vertical axis) as a function of victimization (low, average and high; plotted on the horizontal axis) for different levels of self-defeating humor style use (low, average and high; plotted as separate lines). (**b**) Psychological school adjustment (plotted on the vertical axis) as a function of self-defeating humor style use (low, average and high; plotted on the horizontal axis) for different levels of victimization (low, average and high; plotted as separate lines). *Note.* Data for visualizing the conditional effects were taken from the syntax output of the PROCESS macro 4.00 [52]. Values for victimization (low = 1.00; average = 1.87; high = 3.31); values for self-defeating humor style (low = 1.98; average = 3.23; high = 4.48).

**Figure 5 ijerph-19-11415-f005:**
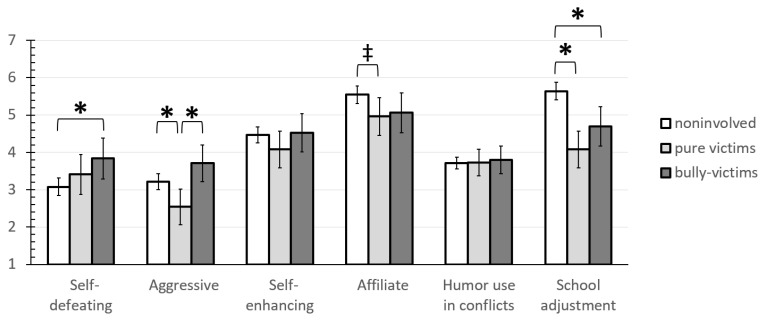
ANCOVA results: Marginal means of bullying-related role groups regarding humor styles and school adjustment. *Note.* Whiskers represent 95% confidence intervals. Results of Tukey post hoc tests: ‡ *p* ≤ 0.10, * *p* ≤ 0.05 (also including significance at lower significance levels).

**Figure 6 ijerph-19-11415-f006:**
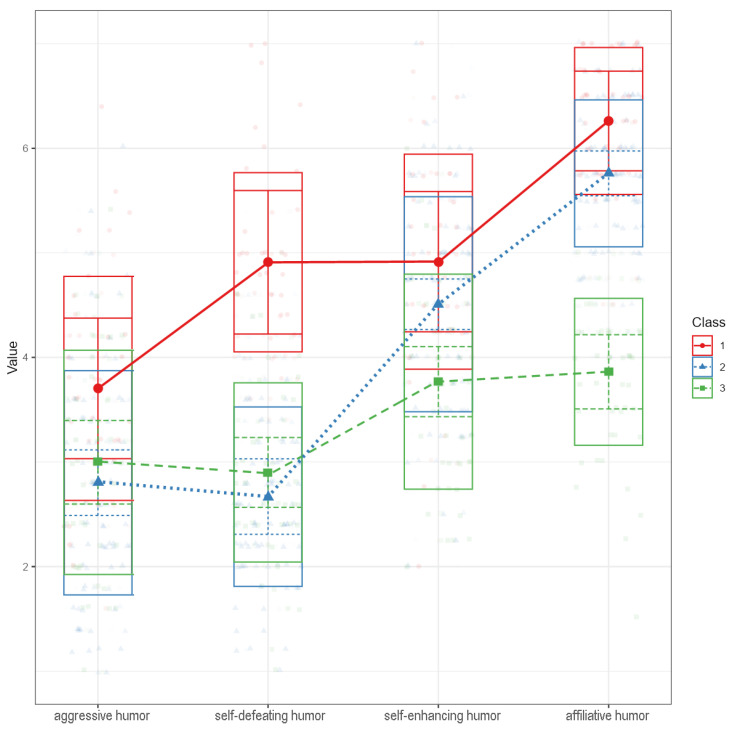
Latent profile plot of the three-class solution of different humor style uses. *Note.* Produced with Jamovi [54].

**Figure 7 ijerph-19-11415-f007:**
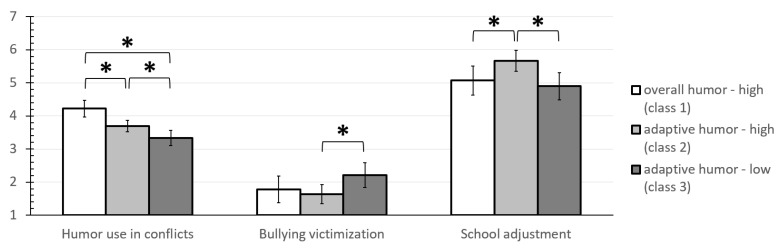
ANCOVA results: Marginal means of humor-related latent profile classes regarding humor use in conflicts, bullying victimization and psychological school adjustment. *Note.* Whiskers represent 95% confidence intervals. Results of Tukey post hoc tests: * *p* ≤ 0.05 (also including significance at lower significance levels).

**Table 1 ijerph-19-11415-t001:** Descriptive statistics and bivariate correlations of the study variables.

Variable	*M*	*SD*	01	02	03	04	05	06	07	08	09	10	11	12
01. Gender (0 = *female*)	0.22	—	—											
02. Age	22.63	2.13	−0.04	—										
03. Class conflict frequency	2.91	0.91	0.03	−0.02	—									
04. Pure victim (0 = *no*)	0.14	—	0.001	−0.11	**0.21 ****	—								
05. Bully-victim (0 = *no*)	0.12	—	−0.02	−0.02	**0.21 ****	−0.15 ‡	—							
06. Victimization	1.89	1.47	−0.03	−0.10	**0.43 *****	**0.61 *****	**0.56 *****	*0.87*						
07. Aggressive humor	3.06	1.13	**0.21 ****	0.03	0.12	**−0.21 ****	**0.21 ****	0.01	*0.71*					
08. Self-defeating humor	3.23	1.25	−0.01	−0.07	**0.27 *****	0.12	**0.22 ****	**0.29 *****	**0.35 *****	*0.75*				
09. Self-enhancing humor	4.39	1.11	0.03	**0.16 ***	−0.10	−0.15 ‡	0.01	−0.08	0.06	**0.21 ****	*0.62*			
10. Affiliative humor	5.34	1.18	0.10	0.001	0.14 ‡	−0.11	−0.06	−0.05	0.13 ‡	**0.26 *****	**0.33 *****	*0.82*		
11. Humor use in conflicts	3.68	0.80	0.07	0.03	0.07	−0.001	0.04	0.07	**0.25 ****	**0.32 *****	**0.51 *****	**0.38 *****	*0.68*	
12. School adjustment	5.24	1.34	0.06	0.09	**−0.37 *****	**−0.41 *****	**−0.24 ****	**−0.57 *****	−0.04	**−0.37 *****	**0.25 *****	0.10	−0.06	*0.93*

*Note.* Significant values (*p* ≤ 0.05) are displayed in bold; if applicable, internal consistency reliabilities (Cronbach alphas) are displayed in italics in the main diagonal. ‡ *p* ≤ 0.10, * *p* ≤ 0.05, ** *p* ≤ 0.01, *** *p* ≤ 0.001.

**Table 2 ijerph-19-11415-t002:** Total, direct, total indirect and indirect effects of the model with victimization mediating the association between humor styles and psychological school adjustment.

							95% CI			
Predictors	Mediator	Outcome	Estimate	*SE*	*z*	*p*	Lower	Upper	*Std* (all)	*Std* (nox)
**Total effects**										
Aggressive humor	—	School adjustment	0.129	0.083	1.545	0.122	−0.035	0.293	0.108	0.108
Affiliative humor	—	School adjustment	**0.172 ***	0.080	2.161	0.031	0.016	0.328	0.151	0.151
Self-enhancing humor	—	School adjustment	**0.301 *****	0.084	3.578	<0.001	0.136	0.467	0.252	0.252
Self-defeating humor	—	School adjustment	**−0.453 *****	0.079	−5.757	<0.001	−0.607	−0.299	−0.423	−0.423
**Direct effects**										
Aggressive humor	—	School adjustment	0.069	0.076	0.906	0.365	−0.080	0.217	0.058	0.058
Affiliative humor	—	School adjustment	0.099	0.072	1.374	0.170	−0.042	0.241	0.087	0.087
Self-enhancing humor	—	School adjustment	**0.289 *****	0.076	3.823	<0.001	0.141	0.438	0.242	0.242
Self-defeating humor	—	School adjustment	**−0.338 *****	0.073	−4.641	<0.001	−0.481	−0.196	−0.316	−0.316
**Indirect effects**										
Aggressive humor	Victimization	School adjustment	0.060	0.038	1.586	0.113	−0.014	0.135	0.051	0.051
Affiliative humor	Victimization	School adjustment	**0.073 ***	0.037	1.968	0.049	0.0003	0.145	0.064	0.064
Self-enhancing humor	Victimization	School adjustment	0.012	0.037	0.326	0.744	−0.061	0.085	0.010	0.010
Self-defeating humor	Victimization	School adjustment	**−0.115 ****	0.039	−2.932	0.003	−0.191	−0.038	−0.107	−0.107

*Note.* Calculated with JASP [51]. Delta method standard errors; full information maximum likelihood estimator. *Std* = Standardized estimates. Significant vales (*p* ≤ 0.05) are displayed in bold. * *p* ≤ 0.05, ** *p* ≤ 0.01, *** *p* ≤ 0.001.

**Table 3 ijerph-19-11415-t003:** The effect of victimization on school adjustment moderated by humor styles while controlling for gender, age and class conflict frequency.

	Effect on School Adjustment	*SE*	*t* Value	*p* Value	Lower Limit 95% CI	Upper Limit 95% CI
Intercept	**5.653 *****	0.734	7.697	<0.001	4.202	7.103
Confounders						
Gender	0.060	0.155	0.391	0.696	−0.245	0.366
Age	−0.001	0.030	−0.024	0.981	−0.061	0.059
Class conflict frequency	−0.104	0.096	−1.079	0.282	−0.295	0.087
Conditional effects						
Victimization	**−0.298 *****	0.068	−4.367	<0.001	−0.432	−0.163
Aggressive humor	0.058	0.065	0.887	0.377	−0.071	0.186
Affiliative humor	**0.166 ***	0.070	2.366	0.019	0.027	0.304
Self-enhancing humor	**0.270 *****	0.072	3.734	<0.001	0.127	0.413
Self-defeating humor	**−0.364 *****	0.065	−5.583	<0.001	−0.493	−0.235
Interaction terms						
Victimization×aggressive humor	**0.145 ***	0.059	2.477	0.014	0.029	0.261
Victimization×affiliative humor	0.065	0.053	1.240	0.217	−0.039	0.170
Victimization×self-enhancing humor	−0.009	0.038	−0.244	0.808	−0.084	0.066
Victimization×self-defeating humor	**−0.145 *****	0.041	−3.562	<0.001	−0.226	−0.065

*Note.* The PROCESS Macro [52] (model 1) was used with psychological school adjustment as dependent variable (*y*), victimization as the focal predictor (*x*), self-defeating humor style as moderator (*W*) and the following covariates: gender, age, class conflict frequency, aggressive humor style, affiliative humor style, self-enhancing humor style, interaction victimization×aggressive humor, interaction victimization×affiliative humor, interaction victimization×self-enhancing humor and interaction victimization×self-defeating humor. All predictors and covariates that define interaction terms were mean-centered to ease interpretability (means of humor styles: aggressive = 3.056; affiliative = 5.336; self-enhancing = 4.390; self-defeating = 3.229; mean of victimization = 1.885). A heteroscedasticity consistent standard error and covariance matrix estimator was used (Huber–White). Significant vales (*p* ≤ 0.05) are displayed in bold. * *p* ≤ 0.05, *** *p* ≤ 0.001.

**Table 4 ijerph-19-11415-t004:** ANCOVA results: Marginal means and standard errors in bullying-related role groups regarding humor-related variables and psychological adjustment in school.

	Noninvolved		Pure Victims		Bully-Victims		ANCOVA Results	
Variables	*M_adj_*	*SE*	*M_adj_*	*SE*	*M_adj_*	*SE*	*F*(2, 161)	η_p_^2^
Aggressive humor style	3.22	0.11	2.54	0.24	3.71	0.25	**6.547 ****	0.075
Affiliate humor style	5.55	0.12	4.96	0.26	5.06	0.27	**3.168 ***	0.038
Self-enhancing humor style	4.47	0.11	4.08	0.25	4.53	0.26	1.241	0.002
Self-defeating humor style	3.08	0.12	3.41	0.27	3.84	0.28	**3.548 ****	0.042
General humor use in conflicts	3.72	0.08	3.73	0.18	3.80	0.19	0.068	0.0008
Psychological school adjustment	5.64	0.12	4.08	0.25	4.70	0.27	**19.641 *****	0.196

*Note.* Calculated with JASP [51]. Marginal mean estimates are adjusted for gender, age and conflict frequency in class. Significant values (*p* ≤ 0.05) are displayed in bold. * *p* ≤ 0.05, ** *p* ≤ 0.01, *** *p* ≤ 0.001.

**Table 5 ijerph-19-11415-t005:** ANCOVA results: Marginal means and standard errors in humor style-related latent profile classes regarding humor-related variables, victimization and psychological adjustment in school.

	Humor Class 1 “Overall High”		Humor Class 2 “Adaptive High”		Humor Class 3 “Adaptive Low”		ANCOVA Results	
**Humor-related variables**	** *M_adj_* **	** *SE* **	** *M_adj_* **	** *SE* **	** *M_adj_* **	** *SE* **	** *F* ** **(2, 162)**	**η_p_^2^**
Aggressive humor style	3.77	0.178	2.87	0.128	3.22	0.168	**9.080 *****	0.101
Self-defeating humor style	4.91	0.131	2.53	0.094	2.80	0.123	**122.429 *****	0.602
Self-enhancing humor style	5.01	0.171	4.51	0.122	3.69	0.161	**18.027 *****	0.182
Affiliate humor style	6.25	0.114	5.81	0.081	3.83	0.107	**169.482 *****	0.677
General humor use in conflicts	4.22	0.125	3.69	0.089	3.33	0.117	**14.694 *****	0.154
**Bullying and school-related variables**	** *M_adj_* **	** *SE* **	** *M_adj_* **	** *SE* **	** *M_adj_* **	** *SE* **	** *F* ** **(2, 161)**	**η_p_^2^**
Bullying victimization	1.78	0.223	1.63	0.160	2.21	0.210	**2.953 ‡**	0.035
Psychological school adjustment	5.07	0.205	5.67	0.147	4.90	0.193	**7.087 *****	0.080

*Note.* Calculated with Jamovi [54]. Marginal mean estimates are adjusted for gender, age and conflict frequency in class. Significant values (*p* ≤ 0.05) are displayed in bold. ‡ *p* ≤ 0.10, *** *p* ≤ 0.001.

## Data Availability

The data presented in this study are available upon request from the corresponding author.

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
