# Peer review of "Humor Styles, Bullying Victimization and Psychological School Adjustment: Mediation, Moderation and Person-Oriented Analyses"

_ijerph, 2022, doi:10.3390/ijerph191811415_

Round 1

Reviewer 1 Report

The article constitutes substantial research. The scope and the research context are well defined and the method is well developed. The results are well presented and connected to previous findings. Though, the introduction section is not well presented. Section 1.3. is missing. Section 1.4. refers to variable-oriented analysis but there is not a clear connection with the victimization. Maybe, the missing section 1.3. could provide the underlying connection. I think that such a connection is important given that the variable-oriented analysis is a vital part of the respective methodology.

Reviewer 2 Report

I am just wondering the statistical methods used for analysis. The auhtors need to provide explanation regarding this. 

What are the main motivation doing this research, this needs to be clearly stated in the introduction. Please also cite the previous research that have been done so the readers can understand the research gap and how important this research is. THis means that the introduction needs to be re written to make it stronger

Reviewer 3 Report

First of all, I consider that the article presents an interesting topic for the scientific world, specifically in the educational field. For what I consider that the article is publishable, once the following nuances have been improved.

Regarding the introduction, it is quite complete and supports the theme to be analyzed, however, the author uses scientific literature from 2003, it is recommended that the citations of more than five years be adjusted to the present, since it will allow the study to be specified to the needs more up-to-date on the research topic.

Regarding the methodology, various software and analyzes are used that give quality and support to the study.

To end the discussion, as well as the introduction, the citations used must be updated so that they adjust to the needs of the current context.

Finally, I would recommend the author to review the English and include a section on the limitations of the study, which will allow for a more complete investigation.

Made these small changes I consider that the article is publishable.

Round 2

Reviewer 2 Report

The authors has responded my concern appropriately. It can be published